# Usefulness of a Fork-Tip Needle in Endoscopic Ultrasound-Guided Fine-Needle Biopsy for Gastric Subepithelial Lesions

**DOI:** 10.3390/diagnostics11101883

**Published:** 2021-10-12

**Authors:** Mika Takasumi, Takuto Hikichi, Minami Hashimoto, Jun Nakamura, Tsunetaka Kato, Ryoichiro Kobashi, Takumi Yanagita, Rei Suzuki, Mitsuru Sugimoto, Yuki Sato, Hiroki Irie, Tadayuki Takagi, Masao Kobayakawa, Yuko Hashimoto, Hiromasa Ohira

**Affiliations:** 1Department of Gastroenterology, Fukushima Medical University School of Medicine, Fukushima 960-1295, Japan; paper@fmu.ac.jp (M.T.); mi-hashi@fmu.ac.jp (M.H.); junn7971@fmu.ac.jp (J.N.); tsune-k@fmu.ac.jp (T.K.); rkobashi@fmu.ac.jp (R.K.); takumi-y@fmu.ac.jp (T.Y.); subaru@fmu.ac.jp (R.S.); kita335@fmu.ac.jp (M.S.); dorcus@fmu.ac.jp (Y.S.); hirokiri@fmu.ac.jp (H.I.); daccho@fmu.ac.jp (T.T.); h-ohira@fmu.ac.jp (H.O.); 2Department of Endoscopy, Fukushima Medical University Hospital, Fukushima 960-1295, Japan; mkobaya@fmu.ac.jp; 3Medical Research Center, Fukushima Medical University, Fukushima 960-1295, Japan; 4Department of Pathology, Fukushima Medical University School of Medicine, Fukushima 960-1295, Japan; ykhykh@fmu.ac.jp

**Keywords:** endoscopic ultrasound, endoscopic ultrasound-guided fine-needle aspiration, endoscopic ultrasound-guided fine-needle biopsy, fork-tip needle, subepithelial lesion

## Abstract

The sample adequacy and diagnostic accuracy of an endoscopic ultrasound (EUS)-guided fine-needle aspiration (EUS-FNA) for gastric subepithelial lesions (SELs) have been reported to be imperfect. To resolve these issues, a fork-tip needle as an EUS-guided fine-needle biopsy (FNB) needle has been developed. This study was conducted to evaluate the usefulness of a fork-tip needle in an EUS-FNB for gastric SELs. Seventy-nine patients who received an EUS-FNA or FNB using a fork-tip needle for gastric SELs were included in the study. The sample adequacy and diagnostic accuracy were compared between the EUS-FNB with the fork-tip needle group (fork-tip group, *n* = 13) and the EUS-FNA with FNA needle group (FNA group, *n* = 66). In addition, a multivariate analysis of the factors influencing diagnostic accuracy was conducted. Regarding sample adequacy, there was no significant difference between the groups (100% vs. 90.9%, respectively; *p* = 0.582). The diagnostic accuracy of the fork-tip group was numerically higher than that of the FNA group (92.3% vs. 81.8%, respectively; *p* = 0.682). In a multivariate analysis, the diagnostic accuracy was related to the tumor size and location of the SEL but not to the needle type. In conclusion, this study does not show statistical superiority, but suggests the useful potential of a fork-tip needle.

## 1. Introduction

Most gastric subepithelial lesions (SELs) are mesenchymal tumors, which require histological immunostaining for diagnosis. Among them, gastrointestinal stromal tumors (GISTs) are especially common and require surgery if diagnosed [1]. However, it is difficult to obtain sufficient tissue with a mucosal biopsy because the surface of the SEL is covered with normal gastric mucosa. Endoscopic ultrasound (EUS)-guided fine-needle aspiration (EUS-FNA) is a conventional method used to obtain specimens from gastric SELs [2]. The diagnostic accuracy of the upper gastrointestinal (GI) SELs by EUS-FNA has been reported to be about 62.0–93.4% [1,2,3,4,5]; this accuracy rate is not high compared to that of pancreatic tumors and lymph node swelling [6].

Recently, endoscopic ultrasound-guided fine-needle biopsy (EUS-FNB) that uses a needle capable of cutting out tissue has been developed [7,8,9,10,11,12,13]. Among the various types of FNB needles, a fork-tip needle has a unique structure with a sharp tip that splits into two halves (Figure 1a), and its usefulness has been reported in pancreatic tumors [14,15,16,17,18,19]. However, few studies have referred to a EUS-FNB for GI SELs using the fork-tip needle. We performed a retrospective comparison study to determine whether an EUS-FNB for gastric SELs using a fork-tip needle could yield more adequate samples and provide a more accurate diagnosis than an EUS-FNA using FNA needles.

## 2. Methods

### 2.1. Study Design and Patients

The aim of this study was to compare the sample adequacy and diagnostic accuracy of an EUS-FNB for gastric SELs using a fork-tip needle with an EUS-FNA using FNA needles. We retrospectively accessed medical records at Fukushima Medical University Hospital, which is one of the largest flagship hospitals in the Fukushima prefecture in Japan. Consecutive patients who underwent an EUS-FNA/B for gastric SELs at the hospital from January 2015 to July 2021 were included in the study. Patients who underwent an EUS-FNB using the fork-tip needle were defined as the “fork-tip group”, and those who underwent an EUS-FNA using FNA needles were defined as the "FNA group".

### 2.2. EUS-FNA/B Indication and Procedure

The indications for an EUS-FNA/B of gastric SELs were as follows: (1) larger than 10 mm in diameter on EUS imaging; (2) hypoechoic lesion suspicious of GIST or SEL-like cancer; (3) located in a puncturable area under EUS; (4) no inevitable blood vessel in the puncture line; and (5) the needle did not extend outside the gastric wall during puncture.

Patients received unconscious sedation and monitored anesthesia care during the procedures. An EUS-FNA/B with rapid on-site evaluation (ROSE) [20] was performed with a linear or convex array of EUS as follows: GF-UCT260, GF-UC240P-AL5, and TGF-UC260J (all EUS scopes were from Olympus Co., Tokyo, Japan). A 22-gage manually operated needle device was usually used for punctures, and a 25G needle was selected depending on the case. For early period cases, conventional FNA needles were used as puncture needles. An FNB needle, called a Franseen needle (Acquire; Boston Scientific Co., Tokyo, Japan), was used as the first choice from 2019, and a fork-tip needle (SharkCore; Covidien Japan Co., Tokyo, Japan) was used as the first choice from August 2020. The details of the proportion of FNA/FNB cases for each year are as follows: 15/0 in 2015; 22/0 in 2016; 22/0 in 2017; 10/0 in 2018; 5/3 in 2019 (three cases of FNB used the Franseen needle); 0/10 in 2020 (three cases of FNB used the Franseen needle and the others used the fork-tip needle); and 0/6 in 2021 (all cases used the fork-tip needle). For the EUS-FNA/B technique, a gastric SEL was punctured by each needle with the stylet and then negative pressure suction was applied after the stylet was removed. Once each needle was passed into the gastric SEL, suction was applied at 20 mL, and the needle was moved back and forth within the SEL about 20 times (Figure 1b). The slow pull method was adapted if the blood flowed into the syringe. After observing the cellular components with ROSE, two more punctures were added to collect specimens for immunostaining histological evaluation. Some of the specimens collected were wet-fixed with Cyto-quick for ROSE, and the others were dry-fixed with Papanicolaou or Giemsa stain. Both these specimens were evaluated using cytological methods. When a sufficient number of specimens were collected with ROSE, the specimens were placed in formalin bottles for histological evaluation (Figure 1c). Formalin-fixed specimens were first evaluated with hematoxylin–eosin (HE) staining. Additional immunostaining was performed, including c-kit, desmin, and S-100, for example, for suspected mesenchymal tumors, and CD markers for suspected lymphomas.

### 2.3. Endpoints of This Study

The primary outcomes were the comparisons of the sample adequacy and diagnostic accuracy of EUS-FNA/B tumors between the fork-tip group and the FNA group. Adequate sampling was defined as adequate cellular components observed by cytological or histological evaluations. When the cellular component was missing, or too small to evaluate, it was considered to be inadequate sampling. Diagnostic accuracy was defined as the percentage of the EUS-FNA/B pathological diagnoses that matched the final diagnoses. The results of the histological evaluations were reflected in the diagnostic accuracy. Final diagnoses were confirmed by surgical specimens if patients received resections. If patients did not receive a resection and were followed up because the results from EUS-FNA/B and other image modalities were benign, final diagnoses were defined by the pathological results from the EUS-FNA/B. Patients with suspected mesenchymal tumors or lymphomas that could not be stained with immunostaining because of insufficient amounts of histological specimens were classified as undiagnosed.

Secondary outcomes were adverse events (AEs) and the factors influencing diagnostic accuracy in all patients. AEs were defined according to the Common Terminology Criteria for Adverse Events ver. 5.0 as follows: bleeding that required endoscopic hemostasis or blood transfusion, perforation with free air confirmed in the image, and infection that required some treatment. The factors to be analyzed were age, sex, tumor size, tumor location (upper, middle, and lower stomach), the number of punctures, and the type of puncture needle (fork-tip group or FNA group). Tumor sizes were quantified using the largest diameter measured during EUS-FNA.

### 2.4. Statistical Analyses

IBM SPSS statistics software version 27 (IBM, Armonk, NY, USA), and GraphPad Prism 6 (GraphPad Software, San Diego, CA, USA), were used for the statistical analyses of the values. The statistical significance of the differences between the groups was determined using a Mann–Whitney U test, a Chi-squared test, or a Fisher’s exact test. Multivariate logistic regression analysis was performed to assess the significant predictors of diagnostic accuracy (correct diagnosis and undiagnosable). A *p* values less than 0.05 were considered statistically significant.

## 3. Results

### 3.1. Patient and Lesion Characteristics

During the study period, a total of 85 patients had an EUS-FNA/B for gastric SELs. Among them, six patients who had an EUS-FNB with the Franseen needle were excluded. Finally, 79 patients were analyzed in this study. Table 1 presents summary information on patient characteristics and the features of the SELs. There were 13 patients in the fork-tip group and 66 patients in the FNA group. There were no significant differences in age, sex, tumor size, needle gage, or tumor location between the two groups. The proportion of patients with surgical resection was significantly higher in the fork-tip group (*p* = 0.019).

The final diagnoses are summarized in Table 2. In the fork-tip group, mesenchymal tumors were the majority (84.6%), and there were ten patients with GIST and one patient with a leiomyoma. The FNA group included forty-one patients with GIST and eight patients with leiomyomas. Ten patients (15.2%) were evaluated as undiagnosed. All endoscopists who performed an EUS-FNA/B had experience with the procedures for more than three years.

### 3.2. Diagnostic Ability and AEs of EUS-FNA/Bs

The results of the EUS-FNA/Bs are revealed in Table 3. The rate of adequate sampling was 100% (13/13) in the fork-tip group, and 90.9% (60/66) in the FNA group. The diagnostic accuracy was 92.3% (12/13) in the fork-tip group, and 81.8% (54/66) in the FNA group, which showed a higher tendency in the fork-tip group, although no significant difference was detected (*p* = 0.682). The patient who was classified as undiagnosed in the fork-tip group had a malignant lymphoma. In this case, lymphocytes were obtained by the EUS-FNB, but neoplastic lymphocytes were not diagnosed based on the HE-stained histological specimen. Furthermore, immunostaining could not be performed because of an insufficient number of specimens. Instead, endoscopic subepithelial dissection was performed for the purpose of diagnosis, and the patient was finally diagnosed as having a diffuse large B-cell lymphoma. In contrast, twelve patients in the FNA group could not be diagnosed. Two of them underwent surgical resection and were diagnosed with a leiomyoma and schwannoma, respectively. Furthermore, although spindle-shaped cells were observed in six patients, they were suspected to be mesenchymal tumors, and immunostaining could not be performed because of an insufficient number of specimens. Three of the remaining four patients also had suspected mesenchymal tumors on EUS imaging, but adequate samples were not available by EUS-FNA for pathological evaluation. These were small tumors (<20 mm), and they were followed up without resection. Regarding AEs, infection in the tumor after EUS-FNB occurred in one patient in the fork-tip group. This was a case of an SEL in the upper stomach, which was diagnosed as GIST after four punctures with EUS-FNB. A few days after the EUS-FNB, a fever of 37 °C and elevated C-reactive protein (CRP) of 1.45 were observed, and CT showed a hypoabsorption zone inside the SEL. On the basis of these findings, intratumoral infection was suspected. The patient was treated with oral antibiotics for about two weeks, and the fever and CRP levels improved.

### 3.3. Factors Influencing Diagnostic Accuracy

Logistic regression analysis showed that the tumor size [adjusted odds ratio (OR), 1.477; 95% confidence interval (CI), 1.114–1.958; *p* = 0.004] and location (upper stomach of adjusted OR, 33.150; 95% CI, 2.321–473.487; *p* = 0.010, and middle stomach of adjusted OR, 14.186; 95% CI, 1.069–188.201; *p* = 0.044) were independent predictors of diagnostic accuracy (Table 4). On the other hand, the type of needle was not a significant factor that influenced diagnostic accuracy (adjusted OR, 3.543; 95% CI, 0.194–64.854; *p* = 0.394).

## 4. Discussion

In this study, an EUS-FNB with the fork-tip needle for gastric SELs was shown to have sufficient sample adequacy and high diagnostic accuracy, but there was no significance between the fork-tip group and the FNA group. In addition, the size and location of the tumor affected the diagnostic accuracy of the EUS-FNA/B for gastric SELs.

It is not easy to obtain a sufficient number of specimens with the EUS-FNA of gastric SELs, and some innovations in the technique have been reported. Among them, the wet suction method, in which the puncture needle is filled with saline, has been reported for pancreatic tumors as a method to maintain suction force [21,22,23]. However, in our previous study, there was no advantage of the wet suction method over the conventional method in EUS-FNA for gastric SELs [24]. Therefore, a method that does not rely on suction seems to be important for reliable tissue collection under EUS for gastric SELs. In recent years, “EUS-FNB” has been developed as a new method of tissue collection under EUS, which is based on the technique of shaving off the tissue by modifying the shape of the needle [7,8,9,10,11,12,13].

Three types of needles are used in EUS-FNB: side-bevel needles, Franseen needles, and fork-tip needles [25]. Among these needles, Franseen and fork-tip needles have attracted much attention in recent years [26], and good sampling adequacy and high diagnostic accuracy have been reported mainly for pancreatic tumors [14,15,16,17,18,19]. EUS-FNB has also been reported to have higher diagnostic accuracy than EUS-FNA in GI SELs [11,27]. However, in these reports, the needle used for EUS-FNB was the Franseen needle or all three types, and none of them focused on the fork-tip needle. Therefore, we switched from the Franseen needle to the fork-tip needle, which was reported to provide sufficient sample adequacy and diagnostic accuracy in EUS-FNB for gastric SELs [28].

In this study, we focused on the fork-tip needle and compared the sample adequacy and diagnostic accuracy of the fork-tip needle with the FNA needle. In the recent meta-analysis comparing the FNB and FNA of SELs, FNB was superior to FNA in all diagnostic outcomes, i.e., the sample adequacy, diagnostic accuracy, histologic core procurement rate, and mean number of needle passes [29]. Sample adequacy and diagnostic accuracy in this study were not statistically significantly higher in the FNB group compared to the FNA group, but these results were considered to be due to the small number of cases. The histologic core procurement rate was not evaluated in this study, but future studies may be carried out on the usefulness of the fork-tip needle. The results show that the fork-tip needle had 100% adequate sampling and 92.3% diagnostic accuracy for EUS-FNB and could yield a correct pathological diagnosis except for one case of malignant lymphoma. In the case of lymphoma, insufficient sampling might be involved, due to the inability of effective needle movement within the tumor because the tumor diameter was 13 mm. All mesenchymal tumors were diagnosed accurately, and pathological evaluation, including immunostaining, was possible with specimens obtained by EUS-FNB. There was a significant difference in the treatment method between the fork-tip group and the FNA group in this study. In other words, there were more follow-up cases in the FNA group. Nine of ten patients with an unknown final diagnosis in the FNA group were suspected of having mesenchymal tumors by EUS-FNA or EUS imaging. However, the patients were followed up to an inaccurate diagnosis. An accurate diagnosis of mesenchymal tumors by EUS-FNB with a fork-tip needle may help to determine appropriate treatments. The ease of puncture with the fork-tip needle for gastric SELs was also superior to that of EUS-FNA needles, although it was not included in this study. In addition, the unique shape of the fork-tip needle, with a structure for cutting out tissue, contributes to sufficient sample adequacy and diagnostic accuracy.

As a subanalysis in this study, factors influencing the diagnostic accuracy of the EUS-FNA/B of gastric SELs were evaluated. In a multivariate analysis, the tumor size and location were the factors influencing the diagnostic accuracy of the EUS-FNA/B of gastric SELs. The high diagnostic accuracy was related to the large size of the tumor and the location of the SEL in the upper and middle stomach. In other words, low diagnostic accuracy was associated with a small tumor size and location in the lower stomach. Previously, it was reported that lesions in the lower stomach were associated with insufficient tissue collection in EUS-FNA [30], which was consistent with our results. On the other hand, the OR of the fork-tip needle to the other needles for diagnostic accuracy was 3.122 (95% CI, 0.138–70.519).

There are several limitations to this study. First, this was a study with a small number of cases in a single institution. In particular, the number of cases of EUS-FNB using the fork-tip needle was small. There is a limit to the number of patients who can undergo EUS-FNA/B for gastric SEL at a single institution. Therefore, a multicenter study is necessary for further validation. Second, the procedure time was not measured. Although procedure time could be an indicator of the superiority of the procedure, it was not assessed in this study. Third, there were more cases with follow-up in the FNA group. Finally, this was a retrospective study and could not objectively evaluate the ease of puncture into the GI SELs.

In conclusion, sample adequacy and diagnostic accuracy had no significant differences between the fork-tip group and the FNA group, but the useful potential of the fork-tip needle in an EUS-FNB for gastric SELs was suggested. EUS-FNB using the fork-tip needle yielded a sufficient number of specimens for immunostaining, particularly in the mesenchymal tumors. The majority of gastric SELs are mesenchymal tumors, and accurate diagnosis by EUS-FNB using the fork-tip needle is useful for determining an appropriate treatment. However, because of the small number of cases in this study, future studies with a large number of cases at multiple institutions are needed.

## Figures and Tables

**Figure 1 diagnostics-11-01883-f001:**
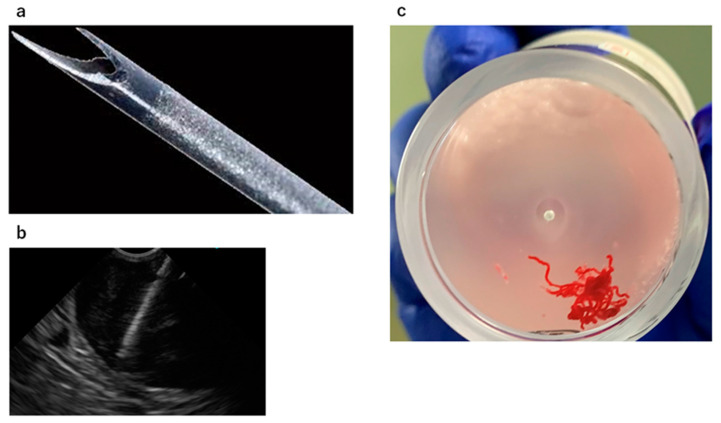
The features of a fork-tip needle. (**a**) The tip of the fork-tip needle is split in two like a shark’s fin. The effect of this structure is to improve the puncture ability to penetrate the stomach wall. (**b**) EUS imaging of a fork-tip needle punctured into a SEL. The needle tip is sandblasted for improved visibility under ultrasound imaging. (**c**) The tissue specimens for histological diagnosis obtained by two punctures of EUS-FNB with the fork-tip needle. Some long and filamentous specimens were also obtained.

**Table 1 diagnostics-11-01883-t001:** Patient characteristics and features of subepithelial lesions.

	Fork-Tip Group (*n* = 13)	FNA Group (*n* = 66)	*p* Value
Age (y) *	68 (41–80)	65 (19–91)	0.449
Sex (male/female)	9/4	29/37	0.131
Tumor size (mm) *	20 (15–95)	20 (8–200)	0.881
Tumor location (U/M/L)	9/4/0	38/18/10	0.322
Needle gage (22G/25G)	13/0	64/2	1.000
Treatment, *n* Resection Chemotherapy Follow-up			0.019 **
Resection	12	33	
Chemotherapy	0	4	
Follow-up	1	29	

* Median (range). ** Statistically significant. FNA, fine-needle aspiration; U, upper stomach; M, middle stomach; L, lower stomach.

**Table 2 diagnostics-11-01883-t002:** Final diagnosis.

	Fork-Tip Group (*n* = 13)	FNA Group (*n* = 66)
GIST	10	41
Leiomyoma	1	8
Schwannoma	0	1
Ectopic pancreas	0	5
Carcinoma	1	1
Malignant lymphoma	1	0
Unknown	0	10

FNA, fine-needle aspiration; GIST, gastrointestinal stromal tumor.

**Table 3 diagnostics-11-01883-t003:** Results of EUS-FNA/B.

	Fork-Tip Group (*n* = 13)	FNA Group (*n* = 66)	*p* Value
Rate of adequate sampling, % (*n*)	100 (13)	90.9 (60)	0.582
Diagnostic accuracy, % (*n*)	92.3 (12)	81.8 (54)	0.682
Number of punctures *	5 (3–9)	5 (1–9)	0.886
Adverse events, % (*n*)	7.7 (1)	0 (0)	0.153

* Median (range); EUS-FNA/B, endoscopic ultrasound-guided fine-needle aspiration or biopsy; FNA, fine-needle aspiration.

**Table 4 diagnostics-11-01883-t004:** Analyses of factors influencing diagnostic accuracy of EUS-FNA/B.

Parameters		Univariate Analysis ^†^	Multivariate Analyses ^‡^	
		*p* Value	Adjusted OR (95% CI)	*p* Value
Age (1-y increments)		0.095	1.035 (0.971–1.102)	0.291
Sex	Female (35)	0.642	1.0 [Reference]	
	Male (44)		0.674 (0.119–3.834)	0.657
Tumor size (1-cm increments)		0.004 *	1.477 (1.114–1.958)	0.007 *
Tumor location	Lower stomach (10) Middle stomach (22) Upper stomach (47)	0.009 *	1.0 [Reference] 14.186 (1.069–188.201) 33.150 (2.321–473.487)	0.044 * 0.010 *
Number of punctures (1-time increments)		0.886	0.729 (0.397–1.338)	0.308
Puncture needle	FNA needle (66)	0.682	1.0 [Reference]	
	Fork-tip needle (13)		3.543 (0.194–64.854)	0.394

^†^ Categorical variables were analyzed with a Chi-squared or Fisher’s exact test, and continuous variables were analyzed with the Mann–Whitney U test. ^‡^ Multinomial logistic regression analysis of all variables. * Statistically significant. EUS-FNA/B, endoscopic ultrasound-guided fine-needle aspiration or biopsy; OR, odds ratio; CI, confidence interval.

## Data Availability

Data available on request because of restrictions, e.g., privacy or ethical.

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
