# Peer review of "Usefulness of a Fork-Tip Needle in Endoscopic Ultrasound-Guided Fine-Needle Biopsy for Gastric Subepithelial Lesions"

_diagnostics, 2021, doi:10.3390/diagnostics11101883_

Round 1
Reviewer 1 Report
The authors presented the results of a retrospective study on EUS guided tissue acquisition in submucosal gastric lesions. The topic is interesting and the paper well written.
However, there are some issues that need to be addressed.
1.First, the FNB group is significantly smaller compared to the FNA group (13 Vs. 66). The authors reported that 6 patients who had FNB with Franseen needle were excluded. I think that a subanalysis including these patients in the FNB group would be interesting to increase the simple size of the FNB group comparing 19 Vs. 66 patients.
2.In Table 3 the authors report one patient with adverse event after FNB with fork tip needle. Please clarify what adverse event was reported.
3. Considering that FNB was recently introduced, please report the proportion FNA/FNB cases for every single year of the 7 included years (2015-2021) This point is optional but I think it would add interesting information for the reader.
Author Response
Authors' comments to Reviewer 1:
The authors presented the results of a retrospective study on EUS guided tissue acquisition in submucosal gastric lesions. The topic is interesting and the paper well written.
→(Response)We would like to express our deepest gratitude to you for reviewing our paper. We are very pleased that you are interested in our paper.
However, there are some issues that need to be addressed.
(Point 1)First, the FNB group is significantly smaller compared to the FNA group (13 Vs. 66). The authors reported that 6 patients who had FNB with Franseen needle were excluded. I think that a subanalysis including these patients in the FNB group would be interesting to increase the simple size of the FNB group comparing 19 Vs. 66 patients.
→(Response)Thank you for helpful suggestion. We were concerned about how to classify the Franseen needles in this study. It is subjective, but we consider that the puncture performance of the Franseen needle is different from that of the fork-tip needle, even if using the same FNB needles. We often experienced difficulty with puncturing the SEL through the GI mucosa using the Franseen needle. Therefore, we excluded the Franssen needle and compared the fork-tip needle with the FNA needles. Actually, there was no significant difference in the primary endpoint when the Franseen needle was included in the FNB needle group. (FNB including Franseen needle vs FNA; diagnostic accuracy 84.2% vs 81.8%, sample adequacy 89.4% vs 90.9%). In this study, although the number of cases is small, we would like to be allowed to compare the fork-tip needle and FNA needle.
(Point 2)In Table 3 the authors report one patient with adverse event after FNB with fork-tip needle. Please clarify what adverse event was reported.
→Thank you for your comment. We described adverse events (AE) in the final part of Results 3.2. We also dded more detail about AE, as they were only briefly described. (Lines 198-203)
(Point 3)Considering that FNB was recently introduced, please report the proportion FNA/FNB cases for every single year of the 7 included years (2015-2021) This point is optional but I think it would add interesting information for the reader.
→(Response)We appreciate your helpful suggestion. We have now added details of the proportion of FNA/FNB cases for each year in the Methods 2.2 section. (Lines 80-83)
Reviewer 2 Report
Abstract. The last phrase of the abstract is not supported by the study’s results. Please rephrase.
Introduction. “swollen lymph nodes”. Please use a more appropriate terminology.
Introduction. Lines 44-46. This phrase is irrelevant. Consider omitting.
Methods. 2.2 The reference to Franseen needle is irrelevant to the study. Please precise the exact periods that FNA and fork-tip FNB were used.
Discussion. The discussion needs further enrichment and results from relevant studies of the literature should be discussed with regard to the current study’s results. Please see a recent meta-analysis that could help you (PMID: 31374187). Moreover, adjust your conclusion based on the results of the study, that is a negative study showing no difference.
Author Response
Authors' comments to Reviewer 2:
We would like to express our deepest gratitude to you for reviewing our paper.
(Point 1)Abstract. The last phrase of the abstract is not supported by the study’s results. Please rephrase.
→(Response)As you mentioned, we have added the fact that this study is a negative study and we rephrased the last phrase of the abstract. (Lines 27-28)
(Point 2)Introduction. “swollen lymph nodes”. Please use a more appropriate terminology.
→(Response)Thank you for your suggestion. We've corrected it to say “lymph node swelling”. (Line 43)
(Point 3)Introduction. Lines 44-46. This phrase is irrelevant. Consider omitting.
→(Response)As you pointed out, the description of Franseen needles was not relevant in this study. This phrase has been deleted.
(Point 4)Methods. 2.2 The reference to Franseen needle is irrelevant to the study. Please precise the exact periods that FNA and fork-tip FNB were used.
→(Response)As you pointed out, the Franseen needle is irrelevant to this study, but we have left the description of Franseen needle to explain the exact periods that FNA and fork-tip FNB were used. We have now added details of the proportion of FNA/FNB cases for each year to the Methods 2.2 section. (Lines 80-83)
(Point 5)Discussion. The discussion needs further enrichment and results from relevant studies of the literature should be discussed with regard to the current study’s results. Please see a recent meta-analysis that could help you (PMID: 31374187). Moreover, adjust your conclusion based on the results of the study, that is a negative study showing no difference.
→(Response)Thank you for your helpful suggestion. We have read and cited the literature you suggested and added some discussion about this study. (Lines 247-254). In addition, we adjusted our conclusion based on the results of the study. (Lines 293-295)
Reviewer 3 Report
In this manuscript, Takasumi et al retrospectively evaluate the usefulness of a fork-tip needle in an EUS-FNB for gastric subepithelial lesions. There were no significant differences in diagnostic accuracy and sample adequacy between fork-tip group and FNA group. Additionally, they found diagnostic accuracy was related to the tumor size and location of the lesions based on the multivariate analysis.
This study potentially showed the usefulness and of a fork-tip needle, compared with EUS-FNA. However, this reviewer recommends that the authors address the following comments.
Major comments:
- Table 1 or Table 3. The author had better to show needle gage in two cohorts. It might associate diagnostic accuracy and adequate sampling rate.
- Table 2. All patients were finally diagnosed in Fork-tip group, although 10 patients were not diagnosed in FNA group. Was it statistically significant? This difference seems to suggest usefulness Fork-tip needle.
- Table 3. The author had better show sampling time in two cohorts.
- Table 4 and Methods section. What is the definition of adverse events? The author should show the number of each adverse event in this study.
- There were no significant differences in diagnostic accuracy and sample adequacy between fork-tip group and FNA group in this study. Then, what is usefulness of fork-tip needle. The authors should describe on this point in Discussion section.
Author Response
Authors' comments to Reviewer 3:
In this manuscript, Takasumi et al retrospectively evaluate the usefulness of a fork-tip needle in an EUS-FNB for gastric subepithelial lesions. There were no significant differences in diagnostic accuracy and sample adequacy between fork-tip group and FNA group. Additionally, they found diagnostic accuracy was related to the tumor size and location of the lesions based on the multivariate analysis.
This study potentially showed the usefulness and of a fork-tip needle, compared with EUS-FNA. However, this reviewer recommends that the authors address the following comments.
→(Response)We would like to express our deepest gratitude to you for reviewing our paper. Furthermore, thank you for your deep understanding and summarization of our paper.
Major comments:
(Point 1)Table 1 or Table 3. The author had better to show needle gage in two cohorts. It might associate diagnostic accuracy and adequate sampling rate.
→(Response)Thank you for your helpful suggestion. As you mentioned, we have now added the needle gage to Table 1. There was no significant difference in needle gages between the two cohorts.
(Point 2)Table 2. All patients were finally diagnosed in Fork-tip group, although 10 patients were not diagnosed in FNA group. Was it statistically significant? This difference seems to suggest usefulness Fork-tip needle.
→(Response)Thank you for your comment and question. Table 2 shows the final diagnosis, which includes cases that were diagnosed by surgery without an accurate diagnosis by the EUS-FNA/B. The diagnostic outcomes of EUS-FNA/B are shown in Table 3. There are 12 cases that were not diagnosed by FNA. We corrected the number of cases that were not diagnosed by FNA in the Result 3.2 section. (Line 189)
(Point 3)Table 3. The author had better show sampling time in two cohorts.
→(Response)Thank you for your helpful suggestion. As you mentioned, our paper would be improved if we showed the procedure times in Table 3. We are sorry, but we did not record the procedure times during EUS-FNA/B. We would like to suggest this as an issue in similar research in the future. We have now added this point to the limitations. (Lines 288-290)
(Point 4)Table 4 and Methods section. What is the definition of adverse events? The author should show the number of each adverse event in this study.
→(Response)Thank you for your comment. As you pointed out, we have now added the definition of adverse events (AE) to the Method 2.3 section. (Lines 122-125). We described the AE in the final part of Results 3.2. We also added a little more detail about AE, as they were only briefly described. (Lines 198-203)
(Point 5)There were no significant differences in diagnostic accuracy and sample adequacy between fork-tip group and FNA group in this study. Then, what is usefulness of fork-tip needle. The authors should describe on this point in Discussion section.
→(Response)Thank you for pointing this out. Based on the results of this study, the usefulness of the fork-tip needle was considered to be its high diagnostic accuracy for mesenchymal tumors. This might have contributed to choosing the treatment. This point was added to the Discussion section. (Lines 261-267)
Round 2
Reviewer 3 Report
In the revised manuscript, all comments have been addressed. Therefore, this reviewer was satisfied for all concerns.